# Preconception health in adolescence and adulthood across generations in the UK: Findings from three British birth cohort studies

Olivia Righton[1]*, Angela Flynn[1,2], Nisreen A. Alwan[3,4], Danielle Schoenaker[5,6,7]

1 Department of Nutritional Sciences, King's College London, London, United Kingdom, 2 School of Population Health, Royal College of Surgeons in Ireland (RCSI), Dublin, Ireland, 3 School of Primary Care, Population Sciences and Medical Education, Faculty of Medicine, University of Southampton, Southampton, United Kingdom, 4 University Hospital Southampton NHS Foundation Trust, Southampton, United Kingdom, 5 NIHR Southampton Biomedical Research Centre, University Hospital Southampton NHS Foundation Trust and University of Southampton, Southampton, United Kingdom, 6 School of Human Development and Health, Faculty of Medicine, University of Southampton, Southampton, United Kingdom, 7 MRC Lifecourse Epidemiology Centre, University of Southampton, Southampton, United Kingdom

* olivia.k.righton@kcl.ac.uk

**Data Availability Statement:** The data used in this study were accessed from the UK Data Service. The datasets related to this research are publicly

## Abstract

Optimising preconception health in women and men holds significant potential for improving pregnancy and offspring health outcomes. To create a picture of the state of preconception health in the UK, this study aimed to describe the prevalence of and changes in preconception health indicators reported in three British birth cohort studies: the 1970 British Birth Cohort Study (BCS70; born in 1970; N = 17,198), Next Steps (1989–1990; N = 15,770), and Millennium Cohort Study (MCS; 2000–2002; N = 19,517). The analysis focused on data obtained during participants' adolescence (16–17 years) and subsequent follow-ups at 25–26 years for BCS70 and Next Steps. Self-reported preconception indicators were defined in line with a previously published review and reported as proportions. Across cohorts, data were available for 14 preconception indicators across four domains: health behaviours and weight, reproductive health and family planning, physical health conditions, and wider determinants of health. However, data for these indicators were not consistently available for all cohort members. Findings suggested persistent suboptimal health behaviours in both genders and across generations, including low intakes of fruit. While alcohol, tobacco, and soft drink intake decreased across generations, obesity prevalence surged. This study underscores the need for public health interventions targeting the root causes of adverse health behaviours towards improvement of fruit consumption, further reduction in alcohol, tobacco, and soft drink consumption, and addressing the escalating obesity rates among individuals of reproductive age. Ongoing monitoring is needed to continue tracking these existing indicators over time, while improved data quality and availability of a wider range of preconception indicators are crucial to comprehensively understanding the complexities of preconception health, enabling the development of more targeted and effective interventions.

available and can be accessed by others in the same manner as the authors. No special access or privileges were granted to the authors, and others can request the data directly through the UK Data Service following their standard access procedures. To access the specific datasets used in this study, users will need to create a free account with the UK Data Service and agree to the terms and conditions for dataset use. Further instructions on dataset access, including terms of use, can be found on the UK Data Service website: (https://ukdataservice.ac.uk). Once registered, the datasets can be found using the following Study Numbers (SN): 1970 British Cohort Study: Age 16 sweep: available via the UK Data Service, SN 6095; Age 26 sweep: available via the UK Data Service, SN 3833; Next Steps: available via the UK Data Service, SN 5545; Millennium Cohort Study: available via the UK Data Service, SN 8682.

**Funding:** DS is supported by the National Institute for Health and Care Research (NIHR) through an NIHR Advanced Fellowship (NIHR302955) and the NIHR Southampton Biomedical Research Centre (NIHR203319)". The funders had no role in study design, data collection and analysis, decision to publish, or preparation of the manuscript.

**Competing interests:** The authors have declared that no competing interests exist.

# Introduction

A life-course approach to improving maternal, paternal, and child health outcomes by optimising parental health before conception, known as preconception health, has attracted interest on a global scale [1]. Such an approach is supported by evidence from developmental biology and epidemiological studies that improving preconception health in women and men represents an opportunity to improve pregnancy outcomes, prevent non-communicable diseases in both parents and their offspring, and thus improve the overall health of two generations at a minimum [2–5].

Recent evidence from national population-based studies suggests most women in the UK enter pregnancy with risk factors for pregnancy and birth complications [6, 7]. For example, among women with an antenatal booking appointment in 2018–19, 73% did not take a folic acid supplement before pregnancy, 22% were living with obesity, and 24% had at least one mental or physical health condition [6]. These risk factors are common even among women actively planning pregnancy [7] and disproportionally affect women from disadvantaged backgrounds [6].

Outside of the UK, a systematic review of 18 studies (n = 16,308) conducted across 10 countries found low adherence to dietary guidelines among women planning pregnancy or already pregnant. Most women did not meet the recommended intakes of vegetables, whole grains, folate, or vitamin E [8]. Among studies focussed on preconception health among women in high-income countries, an analysis of GP records for Australian women aged 18–44 (n = 100) revealed that 14% were smokers, 24% had obesity, 7% had high blood pressure, 5% had diabetes, 28% had a mental health condition, and 17% were prescribed potentially teratogenic medications [9]. Another cross-sectional study of 258 Danish women observed that 43% of women with a high degree of pregnancy planning and 98% of those with low planning did not take folic acid pre-pregnancy. Binge drinking in early pregnancy was reported by 20% of women with high planning and 31% with low planning [10]. In a Dutch cohort (n = 921), 85.5% had a planned pregnancy, of which 69.5% took folic acid, and 50.5% consumed alcohol during pregnancy. Among women in this cohort who believed they were "healthy enough" without preconception care, 9.1% met vegetable intake guidelines, 55.6% drank alcohol during pregnancy, and 30.4% were either overweight or underweight [11].

Current initiatives to optimise and reduce inequalities in preconception health in the UK, among other countries, exist at both the individual and population levels [12]. At the individual level, primary healthcare professionals are, for example, encouraged to assess and manage risk factors in women and couples who are planning a pregnancy [13]. Population-level public health strategies can lead to community-level benefits that include improving preconception health by contributing to reducing health inequalities [14]. Examples include mandatory flour fortification with folic acid [15], the Soft Drinks Industry Levy [16] and calorie labelling legislation [17].

To monitor progress made towards improving preconception health and to inform and evaluate existing and new initiatives, the UK Preconception Partnership has laid out a framework and made recommendations for the annual reporting of preconception health indicators in England [6, 12, 18]. To date, women's preconception health in England has been described in a first report card using antenatal booking (first) appointment data from the National Maternity Services Dataset 2018–19 [6]. Further work is underway to develop a Preconception Health Profile within the UK government Office for Health Improvement and Disparities' existing surveillance platform to support the annual reporting of preconception indicators [6]. Antenatal booking appointment data from the Maternity Services Dataset provides an initial picture of preconception health among women who are pregnant. However, there is currently

no national picture of preconception health among men and women of reproductive age across adolescence and adulthood, irrespective of their pregnancy intention and pregnancy status. Such data would inform, develop, and evaluate health initiatives at key stages prior to pregnancy and track progress made towards improving the health of people who may become pregnant in the future. The present study, therefore, aimed to identify and describe preconception indicators reported among women and men during adolescence and adulthood in three British birth cohort studies (across generations) while also exploring how these indicators changed for cohort members over time.

## Materials and methods

### Study design and population

This study used data from three ongoing longitudinal birth cohort studies, including the 1970 British Birth Cohort Study (BCS70), Next Steps, and the Millennium Cohort Study (MCS) [19–21]. Each of these studies follows large, nationally representative groups of people born in the UK and collects information on health and wellbeing, education, employment, and economic circumstances, among other factors. Data are collected through a combination of phone interviews, face-to-face interviews, self-completion questionnaires, interviewer-administered questionnaires, and parental interviews. The most recent data collection time points (sweeps) for each of these birth cohort studies were among individuals of reproductive age (15–49 years) and, therefore, relevant to contemporary preconception health (Fig 1).

The BCS70 follows a sample of all people (around 17,000) born in England, Scotland, and Wales during one week in 1970 when data were initially collected from mothers and medical records [19]. BCS70 has conducted 11 survey sweeps to date, with the latest sweep conducted in 2021 at age 51. Next Steps, previously known as the Longitudinal Study of Young People in England (LSYPE), follows a sample of around 16,000 people born in England between 1st September 1989 and 31st August 1990 [20]. The study began in 2004 when cohort members were 14 years old. The study was sampled using schools and additionally stratified by deprivation levels of those schools, oversampling more deprived schools and oversampling pupils from minority ethnic groups. Design weights were used to return the cohort to representative proportions of individuals from each ethnic group and deprivation stratum. Cohort members were surveyed yearly until 2010, then in 2015 at age 25, and the latest sweep was conducted in 2022 at age 32. The MCS follows a sample of around 19,000 people born across England,

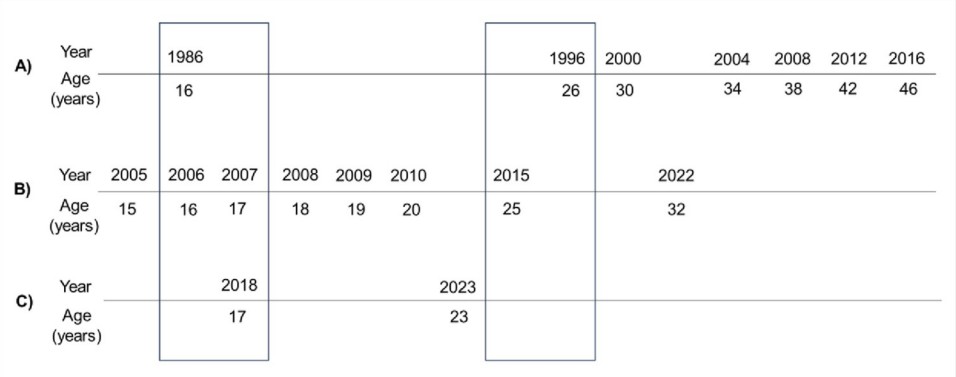

**Fig 1. Overview of data collection sweeps during the reproductive years (15–49 years) in British birth cohort studies. A)** 1970 British Cohort Study; **B)** Next Steps; **C)** Millenium Cohort Study. The two boxes indicate the sweeps used in the current study based on the alignment of age at the time of data collected during the reproductive years.

Scotland, Wales, and Northern Ireland from 2000–2002 [21]. Children from ethnic minorities and disadvantaged backgrounds were oversampled at recruitment, which was accounted for by weighting. There have been seven MCS sweeps, the most recent of which is currently being conducted (2023–24) when cohort members are 23 years of age.

For the present study, to allow for comparison of preconception indicators between cohorts, sweeps were selected for analysis based on the cohort members' age at the time of data collection (i.e., aged 15–49 and comparable age across all three cohorts) (Fig 1). All three studies conducted a sweep when cohort members were aged 16 or 17 and 25 or 26 years, except MCS, which conducted the latest sweep at age 23 in 2023–24.

## Preconception indicators

Preconception indicators for analysis were based on a previous scoping review by Schoenaker et al. [18]. Briefly, this scoping review identified indicators relevant to preconception health based on national and international clinical guidelines and policy documents. Preconception indicators were defined as social, behavioural, or medical risk factors or exposures, as well as their wider determinants, that may impact future pregnancies among all individuals of reproductive age. A total of 66 indicators were identified and grouped into 12 domains, spanning wider determinants of health, healthcare, emotional and social health and support, reproductive health and family planning, health behaviours and weight, immunisations and infections, mental and physical health conditions, medication, and genetic risk.

Data dictionaries for each relevant sweep in the BCS70, Next Steps and MCS were searched to identify which of the 66 preconception indicators were assessed. The process of searching the individual data dictionaries was conducted independently by two researchers (OR and DS) to minimise the risk of error. Relevant variables were selected for inclusion if data could be used to define preconception indicators in line with those proposed in Schoenaker et al. [18], and indicators were reported in at least two sweeps across all included cohorts.

Education and employment data were available at age 16/17 for all three cohorts. However, the age 16/17 sweeps were not included in the analysis due to the school leaving age in the UK being around or after this age and varying between cohorts.

## Statistical analysis

Variables were re-coded in line with preconception indicator definitions proposed in Schoenaker et al. [18] (S1 Table). The number of participants and percentages were calculated for each preconception indicator, stratified by sex (female and male). Most indicators were relevant to females only based on reviewing clinical guidelines and policies, but some were also relevant to males, such as smoking, obesity, and diabetes [18]. Chi-squared tests were used to compare the prevalence of indicators between 1) sweeps at age 16/17 and age 25/26 within a cohort (change across age), and 2) sweeps that included participants of a similar age across cohorts (change across generations), where relevant data were available. All statistical analyses were conducted using *Stata* version 17.0.

## Ethical approval

Ethical approval was secured by the Centre for Longitudinal Studies from the National Health Service (NHS) Research Ethics Committee, and all participants gave informed consent [19–21]. The BCS70, Next Steps and MCS datasets were accessed via the UK Data Service [22]. Access requires registration with the UK Data Service and acceptance of their End User Licence (EUL). All applicable conditions were complied with in this study.

## Results

Data were available on 14 preconception indicators (25 indicator measures, e.g., underweight, overweight, and obesity as measures within the 'weight' indicator) across four domains: wider determinants of health, reproductive health and family planning, health behaviours and weight, and physical health conditions (Table 1). Data on other preconception indicators were not reported in any of the cohorts (e.g., folic acid supplementation or cervical screening) or were not consistently assessed in at least two sweeps across all included cohorts (e.g., mental health condition or contraception) (S2 Table). At age 16, the maximum sample sizes were 11,615 for BCS70, 12,439 for Next Steps, and 10,757 for MCS-17y. At age 26, BCS70 had a maximum sample size of 9,003, and at age 25, Next Steps had 7,707 participants. These numbers reflect the presence of missing data, given the initial sample sizes of around 17,000, 16,000, and 19,000 for BCS70, Next Steps, and MCS-17y, respectively. Each sweep included approximately 50% female and male participants.

### Wider determinants of health

**Ethnicity.** The ethnicity indicator, defined as the "percentage of women from a minority ethnic group", was assessed in BCS70-16y, BCS70-26y, Next Steps-16y, and Next Steps-25y. In BCS70-16y, 6.5% of participants were from an ethnic minority group, slightly lower than 4.8% in BCS70-26y. In Next Steps-16y and Next Steps-26y, 33.6% and 31.4% of participants were from an ethnic minority group (Table 1).

**Education.** The education indicator, defined as the "percentage of women who had not completed high school education" (i.e., before completing A Levels or equivalent in the UK system), was reported in BCS70-26y and Next Steps-25y. In BCS70-26y, 58.9% of participants had not completed high school, compared to a significantly lower 55.2% in Next Steps-26y ($p<0.0001$) (Table 1).

**Employment.** The employment indicator, defined as the "percentage of women unemployed and seeking work", was reported in BCS70-26y and Next Steps-25y. In BCS70-26y, 5.2% of participants were unemployed and seeking work, compared to 6.2% in Next Steps-26y ($p = 0.15$) (Table 1).

**Housing.** The housing indicator, originally defined as the "percentage of women who do not have access to safe, comfortable, affordable housing", was assessed in Next Steps-25y and MCS-17y. Among Next Steps-25y participants, 1.4% reported living in "sheltered housing" or "a hostel for homeless, refuge, YMCA, etc." and <1.0% of MCS-17y participants reported being "homeless" (Table 1).

**Language.** The language indicator, defined as the "percentage of women speaking a language other than English in the home", was also assessed in BCS70-16y and MCS-17y. In BCS70-16y, 4.6% of participants reported usually speaking a language other than English at home, compared to 1.1% in MCS-17y ($p<0.0001$) (Table 1).

### Reproductive health and family planning

**Obstetric history.** The obstetric history indicator, defined as the "percentage of women (who have previously been pregnant) with a previous miscarriage, termination, or stillbirth", was assessed in Next Steps-25y and MCS-17y. In Next Steps-25y, 22.5% of participants reported that they had experienced such outcomes, compared to 69.6% of MCS-17y participants (Table 1).

### Health behaviours and weight

**Dietary intake.** The dietary intake indicator, originally defined as the "percentage of women/men not consuming a healthy diet in line with the national recommendations", was

**Table 1. Prevalence of preconception indicators and changes across three British birth cohorts.**

| Preconception indicator | BCS70 | | Next Steps | | MCS | | |
| --- | --- | --- | --- | --- | --- | --- | --- |
| | Age 16 n (%) | Age 26 n (%) | Age 16/17 n (%) | Age 25 n (%) | Age 17 n (%) | p value comparing cohorts at age 16/17 | p value comparing cohorts at age 26/27 |
| *Wider determinants of health* | | | | | | | |
| Ethnic minority[a] | N = 3,068 | N = 2,328 | N = 5,753 | N = 5,598 | | | |
| | 198 (6.5) | 111 (4.8) | 1,930 (33.6) | 1,757 (31.4) | | <0.0001 | <0.0001 |
| Not completed high school education[a] | | N = 4,831 | | N = 4,281 | | | |
| | | 2,843 (58.9) | | 2,382 (55.2) | | | <0.0001 |
| Unemployed and seeking work[a] | | N = 1,578 | | N = 4,266 | | | |
| | | 82 (5.2) | | 265 (6.2) | | | 0.15 |
| Homeless[a] | | | | N = 4,269 | N = 3,913 | | |
| | | | | 59 (1.4) | (<1%) | | |
| Language other than English usually spoken at home[a] | N = 4,775 | | | | N = 2,623 | | |
| | 220 (4.6) | | | | 29 (1.1) | <0.0001 | |
| *Reproductive health and family planning* | | | | | | | |
| Previous pregnancy loss[a] | | | | N = 3,099 | N = 79 | | |
| | | | | 697 (22.5) | 55 (69.6) | | |
| *Health behaviours and weight* | | | | | | | |
| Fruit not consumed daily[a] | N = 3,004 | | | | N = 2.638 | | |
| | 2,089 (69.5) | | | | 1,950 (73.9) | <0.0001 | |
| Fruit not consumed daily[b] | N = 2,204 | | | | N = 2,312 | | |
| | 1,683 (76.4) | | | | 1,730 (74.8) | 0.23 | |
| Soft drink consumption >1 serve/day[a] | N = 2,981 | | | N = 4,170 | N = 2,629 | | |
| | 709 (23.8) | | | 893 (21.4) | 478 (18.2) | <0.0001 | |
| Soft drink consumption >1 serve/day[b] | N = 2,184 | | | N = 3,333 | N = 2,308 | | |
| | 673 (30.8) | | | 914 (27.4) | 429 (18.6) | <0.0001 | |
| Underweight (BMI <18.5 kg/m$^2$)[a][e] | N = 3,015 | | | N = 4,281 | N = 3,682 | | |
| | 327 (10.9) | | | 180 (4.2) | 248 (6.7) | <0.0001 | |
| Overweight (BMI 25.0–29.9 kg/m$^2$)[a] | N = 3,015 | | | N = 4,281 | N = 3,682 | | |
| | 387 (13.2) | | | 861 (20.1) | 751 (20.4) | <0.0001 | |
| Overweight (BMI 25.0–29.9 kg/m$^2$)[b] | N = 2,848 | | | N = 3,426 | N = 3,251 | | |
| | 297 (10.4) | | | 1,004 (29.3) | 729 (22.4) | <0.0001 | |
| Obesity (BMI ≥30 kg/m$^2$)[a] | N = 3,015 | | | N = 4,281 | N = 3,682 | | |
| | 71 (2.4) | | | 1,044 (24.4) | 422 (11.5) | <0.0001 | |
| Obesity (BMI ≥30 kg/m$^2$)[b] | N = 2,848 | | | N = 3,426 | N = 3,251 | | |
| | 59 (2.1) | | | 636 (18.6) | 398 (12.2) | <0.0001 | |
| Smoking[a] | N = 3,123 | N = 4,849 | N = 5,932 | N = 4,130 | N = 3,863 | | |
| | 717 (23.0) | 1,743 (36.0) | 1,605 (27.1) | 950 (23.0) | 745 (19.3) | <0.0001 | <0.0001 |
| Smoking[b] | N = 2,327 | N = 4,043 | N = 6,054 | N = 3,291 | N = 3,383 | | |
| | 464 (19.9) | 1,585 (39.2) | 1,245 (20.6) | 963 (29.3) | 648 (19.2) | 0.26 | <0.0001 |

(*Continued*)

**Table 1.** (Continued)

| Preconception indicator | BCS70 | | Next Steps | | MCS | | |
| --- | --- | --- | --- | --- | --- | --- | --- |
| | Age 16 n (%) | Age 26 n (%) | Age 16/17 n (%) | Age 25 n (%) | Age 17 n (%) | p value comparing cohorts at age 16/17 | p value comparing cohorts at age 26/27 |
| Any alcohol consumption[a] | N = 3,512 | N = 4,833 | N = 5,919 | N = 4,134 | N = 3,876 | | |
| | 3,201 (91.1) | 4,618 (95.6) | 3,975 (67.2) | 3,082 (74.6) | 3,010 (77.7) | <0.0001 | <0.0001 |
| Any alcohol consumption[b] | N = 2,569 | N = 4,044 | N = 6,044 | N = 3,294 | N = 3,396 | | |
| | 2,340 (91.1) | 3,903 (96.5) | 4,040 (66.8) | 2,633 (79.9) | 2,623 (77.2) | <0.0001 | <0.0001 |
| *Physical health conditions* | | | | | | | |
| Epilepsy[a] | | N = 4,901 | | | N = 3,973 | | |
| | | <1% | | | <1% | | |
| Epilepsy[b] | | N = 4,102 | | | N = 3,491 | | |
| | | <1% | | | <1% | | |
| Diabetes mellitus[a] | | N = 4,901 | | N = 830 | N = 3,973 | | |
| | | <1% | | <1% | <1% | | |
| Diabetes mellitus[b] | | N = 4,102 | | N = 529 | N = 3,491 | | |
| | | <1% | | <1% | <1% | | |
| Asthma[a] | N = 3,026 | N = 4,901 | | | | | |
| | 110 (3.6) | 439 (9.0) | | | | | |
| History of cancer[b] | | N = 4,901 | | N = 830 | | | |
| | | <1% | | <1% | | | |

For each preconception indicator, the total sample size (N) is reported, as well as the n (%) for participants categorised as 'yes' for each indicator.

P values are provided for variables with two or more available data points for each indicator.

[a] Female participants

[b] Male participants

indirectly assessed in BCS70-16y, Next Steps-25y and MCS-17y (Table 1). Two of these sweeps assessed fruit consumption (BCS70-16y and MCS-17y), and three assessed soft drink consumption (BCS70-16y, Next Steps-25y and MCS-17y) that could be compared to the UK recommendations [23].

At age 16/17, 69.5% and 73.9% of female participants in BCS70-16y and MCS-17y did not consume fruit daily, respectively ($p<0.0001$). Moreover, in BCS70-16y, 23.8% of female participants consumed more than one glass of soft drink per day compared to 18.2% of MCS-17y female participants ($p<0.0001$). Among male participants, there was no significant difference in the proportion who did not consume fruit daily (76.4% in BCS70-16y and 74.8% in MCS-17y, p = 0.23), but similar to female participants, consuming more than one glass of soft drink per day was more common among male participants in BCS70-16y (30.8%) compared to MCS-17y (18.6%) ($p<0.0001$). At age 25, excess soft drink consumption was 21.4% among female participants and 27.4% among male participants in Next Steps-25y.

**Weight.** The weight indicator, defined as the "percentage of women in the underweight, and women/men in the overweight and obesity BMI categories", was assessed in BCS70-16y, Next Steps-25y and MCS-17y.

At age 16/17, the proportion of female participants with underweight (10.9% and 6.7%), overweight (13.2% and 20.4%) and obesity (2.4% and 11.5%) was significantly different in BCS70-16y and MCS-17y, respectively ($p<0.0001$). A similar pattern of results was observed for male participants, with the proportions increasing across subsequent cohorts for

overweight (10.4% and 22.4% in BCS70-16y and MCS-17y, respectively) and obesity (2.1% and 12.2%) ($p<0.0001$). In Next Steps-25y, the proportion of female participants with overweight was 20.1%, and 24.4% had obesity. These proportions were 29.3% and 18.6% for overweight and obesity among male participants, respectively (Table 1).

**Tobacco use.** The tobacco use indicator, defined as the "percentage of women/men who currently smoke", was assessed in all included sweeps. Among female participants, the proportion of those aged 16/17 who smoked was significantly different across cohorts: 23.0% in BCS70-16y, 27.1% in Next Steps-16y and 19.3% in MCS-17y ($p<0.0001$). These differences were not observed for male participants who smoked at age 16/17: 19.9% in BCS70-16y, 20.6% in Next Steps-16y and 19.2% in MCS17y ($p = 0.26$). At age 25/26, the proportion of both female and male participants who smoked was significantly lower in Next Steps-25y (23.0% and 29.3%, respectively) compared with BCS70-26y (36.0% and 39.2%) ($p<0.0001$).

These study populations were restricted to those with data at age 16/17 and age 25/26 for BCS70 and Next Steps to show that the proportions of female and male participants who smoked at age 25/26 were significantly higher (30.7% and 31.8% in BCS70 and 22.7% and 29.3% in Next Steps, respectively) compared with the same participants at age 16/17 (21.8% and 18.2% in BCS70 and 24.6% and 16.7% in Next Steps) ($p<0.0001$) (Table 2).

**Alcohol consumption.** The alcohol consumption indicator, defined as the "percentage of women/men who consume any alcohol", was assessed in all included sweeps. Among female participants, the proportion of those aged 16/17 who consumed any alcohol was significantly different across cohorts: 91.1% in BCS70-16y, 67.2% in Next Steps-16y and 77.7% in MCS-17y ($p<0.0001$). Alcohol consumption among male participants at age 16/17 followed a similar pattern: 91.1% in BCS70-16y, 66.8% in Next Steps-16y and 77.2% in MCS17y ($p<0.0001$). At age 25/26, the proportion of both female and male participants who consumed any alcohol was significantly lower in Next Steps-25y (74.6% and 79.9%, respectively) compared with BCS70-26y (95.6% and 96.5%) ($p<0.0001$) (Table 1).

When restricting these study populations to those with data at age 16/17 and age 25/26 for BCS70 and Next Steps, proportions of female and male participants who consumed any

**Table 2. Prevalence of preconception indicators and changes across age in three British birth cohorts.**

| | BCS70 | | | Next Steps | | |
|---|---|---|---|---|---|---|
| Preconception indicator | Age 16 n (%) | Age 26 n (%) | p value | Age 16/17 n (%) | Age 25 n (%) | p value |
| *Health behaviours and weight* | | | | | | |
| Smoking[a] | N = 2,361 | N = 2,361 | | N = 3,531 | N = 3,531 | |
| | 515 (21.8) | 724 (30.7) | <0.0001 | 870 (24.6) | 802 (22.7) | <0.0001 |
| Smoking[b] | N = 1,487 | N = 1,487 | | N = 2,833 | N = 2,833 | |
| | 270 (18.2) | 473 (31.8) | <0.0001 | 473 (16.7) | 829 (29.3) | <0.0001 |
| Any alcohol consumption[a] | N = 2,630 | N = 2,630 | | N = 3,529 | N = 3,529 | |
| | 2,413 (91.8) | 2,525 (96.0) | <0.0001 | 2,442 (69.2) | 2,661 (75.4) | <0.0001 |
| Any alcohol consumption[b] | N = 1,670 | N = 1,670 | | N = 2,823 | N = 2,823 | |
| | 1,545 (92.5) | 1,619 (97.0) | <0.0001 | 1,897 (67.2) | 2,296 (81.3) | <0.0001 |
| *Physical health conditions* | | | | | | |
| Asthma[a] | N = 2,183 | N = 2,183 | | | | |
| | 80 (3.7) | 182 (8.3) | <0.0001 | | | |

For each preconception indicator, the total sample size (N) is reported, as well as the n (%) for participants categorised as 'yes' for each indicator.

For this analysis comparing the prevalence of an indicator across sweeps within a cohort, the sample size is restricted to participants with valid data at both sweeps.

[a] Female participants

[b] Male participants

alcohol at age 25/26 was slightly higher (96.0% and 97.0% in BCS70 and 75.4% and 81.3% in Next Steps, respectively) compared with the same participants at age 16/17 (91.8% and 92.5% in BCS70 and 69.2% and 67.2% in Next Steps, respectively) ($p < 0.0001$) (Table 2).

**Physical health conditions.** In terms of physical health conditions, epilepsy, diabetes mellitus, asthma, and cancer were assessed in at least two sweeps across all included cohorts (Table 1). The proportion of participants with epilepsy (female and male; BCS70-26y, MCS-17y), diabetes mellitus (female and male; BCS70-26y, Next Steps-25y, MCS-17y) and cancer (female; Next Steps-25y, BCS70-26y) was <1.0% at each sweep. The asthma indicator, originally defined as the "percentage of women with (uncontrolled or unreviewed) asthma", was assessed in BCS70-16y and BCS70-26y, with the proportion of female participants reporting asthma significantly increased between these time points from 3.7% to 8.3% ($p < 0.0001$) (Table 2).

## Discussion

This study aimed to identify and describe preconception indicators in men and women across adolescence and adulthood in three British birth cohort studies, examining changes over time across age and between generations. Data were collected from individuals born in the UK at ages 16/17 and 25/26 years. Data were available on 14 preconception indicators across four domains: health behaviours and weight, reproductive health and family planning, physical health conditions, and wider determinants of health.

The study noted persistent suboptimal preconception health behaviours across generations during adolescence and adulthood, affecting both females and males. The prevalence of overweight and obesity displayed an upward trajectory over time, with adolescents in the MCS-17y cohort being around five times (females) to six times (males) more likely to live with obesity when compared to their counterparts in the BCS70-16y cohort. In the Next Steps-25y cohort, 24.4% of female adults were living with obesity, which is in line with the 22% obesity rate reported in a prior study involving women attending antenatal booking appointments in 2018–19 [6]. Moreover, this study's findings align with existing literature indicating a rising trend in obesity among UK adolescents and young adults across generations [5] and underscores the urgent need for preconception obesity prevention and weight management to mitigate pregnancy-related complications associated with elevated BMI, such as gestational diabetes, preeclampsia, and intrauterine death [24].

Suboptimal dietary intakes were observed across all three cohorts in females and males. Most female and male adolescents in BCS70-16y (69.5% and 76.4%, respectively) and MCS-17y (73.9% and 74.8%) did not consume a portion of fruit daily. This aligns with prior studies, including the UK National Diet and Nutrition Survey (2019), which found that 76% of women and men aged 19–64 consumed fewer than five portions of fruit and vegetables daily [25]. Additionally, most women planning pregnancy did not meet the recommended intake of vegetables in a systematic review of 18 studies (n = 16,308) conducted across 10 countries [8]. Among 921 Dutch women who believed they were "healthy enough" without preconception care, only 9.1% met vegetable intake guidelines [11]. Together, these findings suggest that more attention is needed at a population level to encourage fruit and vegetable consumption in those of reproductive age, for example, through increasing awareness about the nutritional value of fruit and vegetables, emphasising their role in providing nutrients which are vital for overall health and are especially important to support sperm and egg quality and contribute to improved preconception health [26].

Sugar-sweetened soft drinks are classified under "foods high in sugar and fat" in The Eatwell Guide [23], which are recommended to be consumed "less often [than daily] and in smaller amounts". High weekly intakes were noted among adolescents in both BCS70-16y and

MCS-17y cohorts, with between 18.2% (females, MCS017y) and 30.8% (males, BCS70-16y) of participants drinking at least one serving per day. However, a significant decline in intakes between cohorts was observed, potentially indicative of heightened public awareness following the introduction of the Soft Drinks Industry Levy in April 2018 [16] and an increased choice of alternatives in the marketplace, such as flavoured waters and sugar-free soft drinks.

Similarly, a decline was observed in any alcohol consumption between cohorts. However, in all five cohorts, the majority of women and girls reported alcohol use, raising concerns due to the association between maternal preconception exposure to alcohol and adverse perinatal outcomes independent of gestational exposure [27]. Previous studies support this, with 55.6% of Dutch women who believed they were "healthy enough" without preconception care drinking alcohol (n = 921) and 20%-31% of Danish women (n = 258) reporting binge drinking during early pregnancy [10, 11]. Most men and boys in all five sweeps reported drinking some alcohol, albeit with a discernible decline among female and male adolescents and adults across generations. Moreover, national data indicate that the amount drinkers report consuming has fallen in the UK since 2005, particularly among younger drinkers [28].

A significant decrease in smoking prevalence was noted among both female and male adults between BCS70-26y and Next Steps-25y. This finding aligns with existing data indicating that the proportion of those aged 16 and over smoking tobacco in Great Britain has declined since 1974, when national surveys on smoking began [29]. When the study populations were restricted to those with data at age 16/17 and age 25/26 for BCS70 and Next Steps, a higher percentage of both females and males reported smoking at age 25/26 (30.7% and 31.8%, respectively) compared to when they were 16/17 (21.8% and 18.2% respectively), emphasising the importance of ongoing public health campaigns and policies to discourage tobacco consumption among those of reproductive age. Insufficient research exists on the safety of e-cigarettes during the preconception period and pregnancy [30]. More data are needed to comprehensively evaluate vaping behaviours among those of reproductive age to foster a nuanced understanding of the associated health implications. Since almost half of all pregnancies are unplanned [31], many women of reproductive age will become pregnant without realising it. Thus, preconception and early prenatal exposure to smoking and vaping are likely to impact pregnancy and offspring outcomes.

This study observed a consistently low prevalence (<1%) of physical health conditions, including epilepsy, diabetes mellitus, asthma, and cancer, across sweeps. However, existing evidence shows a rapid rise in pregestational type 2 diabetes mellitus between 1995 and 2012 in the UK [32]. Moreover, Northern Europe has reported increases of 33–44% in pregnancies complicated by type 1 diabetes and increases of 90–111% in pregnancies complicated by type 2 diabetes have been reported in Northern Europe [33, 34]. These data highlight the importance of ongoing monitoring of physical health conditions due to their potential impact on pregnancy outcomes and the need for specialised care pathways [35].

Regarding wider determinants of health, this study noted varying proportions of ethnic minority study participants across the cohorts, reflecting deliberate oversampling of ethnic minority groups (Indian, Pakistani, Bangladeshi, Black African, Black Caribbean, and Mixed) to ensure adequate sample sizes [36]. UK census data show an increase in the proportion of the population from an ethnic minority background (from 13.8% in 2011 to 18.0% in 2021) [37]. While we recognise that ethnicity as a preconception indicator is not modifiable, ethnic inequalities in health have been linked to wider social, economic, and environmental factors [38]. Public health interventions addressing such (intersectional) inequalities in ethnic minority communities are therefore highly relevant to improving population-level preconception health [39, 40]. Educational differences were also evident, with slightly fewer BCS70-16y females completing high school compared to Next Steps-26y (58.9% vs. 55.2%, respectively), in

line with trends observed in UK census data [41]. These findings highlight the importance of tailoring public health interventions and healthcare to changing social and cultural contexts to ensure equitable reproductive health outcomes.

## Strengths and limitations

This study provides a picture of preconception health over three generations, strengthened by the large-scale national datasets and reflecting the changing demographics in the UK, particularly the increasing representation of ethnic minority groups in later cohorts. However, it is not without limitations. Foremost among these limitations is the substantial proportion of preconception indicators (52 out of 66) that were either inconsistently reported across cohorts due to variance in definitions or questions or not collected entirely. The unavailability of data curtailed both the inter-cohort comparability and impeded the study's ability to paint a full picture of preconception health. The inconsistent assessment of mental health across cohorts represents a notable limitation of the datasets. Specifically, the lack of data pertaining to mental health disorders and associated prescribed medications presents a significant drawback, given the potential for teratogenicity.

Additionally, none of the datasets in this study assessed intention to conceive, which is an especially important indicator in the context of preconception health and its heightened relevance to those planning pregnancy. Moreover, even though data were available on some health behaviours such as diet, alcohol consumption and smoking, data on physical activity were not or inconsistently recorded across the sweeps. Pregnancy intention and physical activity data are also not routinely collected in healthcare datasets [6, 18], highlighting a gap in national surveillance for these key preconception indicators.

Secondly, missing data was a limitation of this study, as not all cohort members answered all questions. For example, only 44.8% of BCS70-16y cohort members (5,208 out of 11,622) reported fruit intake. The potential non-random nature of this missing data raises concerns about possible bias, cautioning the interpretation of findings. Furthermore, although common in longitudinal studies, the loss-to-follow-up may mean the cohort samples were not as representative of the UK population in later sweeps as in earlier sweeps, depending on which individuals dropped out, again emphasising the need for careful result interpretation.

Reliance on self-reported data may also have introduced bias, as participants may have provided subjective or socially desirable responses, affecting the accuracy of the findings. Finally, the study was limited by a lack of indicator detail. Limited detail in distinguishing previous pregnancy loss may explain the observed disparity between women from Next Steps-25y and girls from MCS-17y who reported pregnancy loss (22.5% and 69.6%, respectively). Specifically, participants were asked whether those who had been pregnant had experienced a termination, miscarriage, or stillbirth. It is possible that adolescents in MCS-17y had recent experience of pregnancy termination, while adult women in Next Steps-25y may not have reported these as such, especially in cases of past unintended teenage pregnancy.

Moreover, dietary measures didn't align precisely with the categories in The Eatwell Guide [23]. For instance, "take-aways" (Next Steps-25y) and "fast food" (MCS-17y) frequencies might not truly represent habitual consumption of "foods high in sugar and fat", which is why we couldn't include this in the analysis. Additionally, cross-cohort comparisons were restricted to fresh fruit and soft drink intake, which may not adequately reflect overall diet quality.

## Implications

Despite its limitations, this study's findings contribute to understanding the current state of preconception health among adolescent and adult women and men in the UK and how this

has changed across generations. The findings highlight the need to identify and support individuals who are planning pregnancy, for example, through healthcare interactions, but also the need for public health strategies that reach everyone, irrespective of pregnancy intention, from as early as adolescence to reduce common risk factors that are often shaped by structural social, economic, and environmental determinants. The findings also call for standardised questions and data collection on key public health measures across regular and more recent sweeps and birth cohorts to support ongoing monitoring of preconception health indicators and inform policy. Compared with routinely collected health(care) data, survey data from the British birth cohorts provide unique new information on important preconception indicators such as dietary intake, alcohol consumption, education and housing. Collectively, this could be used alongside preconception indicators from other data sources to track progress towards optimising preconception health across critical life stages in women and men who may become pregnant and inform the need for and development of new strategies.

## Conclusion

The findings of this study add to a body of research suggesting that, while trends in preconception tobacco and soft drink consumption have decreased in the UK, overweight and obesity have increased, and fresh fruit consumption remains low. Since almost half of all pregnancies are unplanned, many women of reproductive age will become pregnant without realising it. Thus, the suboptimal health behaviours observed in this study may impact pregnancy and offspring outcomes through preconception and early prenatal exposure. These findings provide essential benchmarks for designing targeted interventions and shaping public health policies in the UK and other similar countries to address obesity, improve dietary patterns, reduce alcohol and tobacco consumption further, and ensure the overall well-being of the preconception population.

## Supporting information

**S1 Table. Overview and definitions of preconception indicators consistently recorded in at least two sweeps at age 16/17 and/or age 25/26 years in the 1970 British Birth Cohort Study (BCS70), Next Steps and Millennium Cohort Study (MCS).**
(DOCX)

**S2 Table. List of preconception indicators not consistently recorded in at least two sweeps at age 16/17 and/or age 25/26 years in the 1970 British Birth Cohort Study (BCS70), Next Steps and Millennium Cohort Study (MCS).**
(DOCX)

## Author Contributions

**Conceptualization:** Danielle Schoenaker.

**Formal analysis:** Olivia Righton, Danielle Schoenaker.

**Methodology:** Olivia Righton, Angela Flynn, Danielle Schoenaker.

**Writing – original draft:** Olivia Righton.

**Writing – review & editing:** Angela Flynn, Nisreen A. Alwan, Danielle Schoenaker.

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
