## [Decision Letter · Decision Letter 0]

9 Apr 2024

PONE-D-23-42108Preconception health in adolescence and adulthood across generations in the UK: findings from three British birth cohort studiesPLOS ONE

Dear Dr. Righton,

Thank you for submitting your manuscript to PLOS ONE. After careful consideration, we feel that it has merit but does not fully meet PLOS ONE’s publication criteria as it currently stands. Therefore, we invite you to submit a revised version of the manuscript that addresses the points raised during the review process.

We look forward to receiving your revised manuscript.

Kind regards,

Katriina Heikkila, PhD

Academic Editor

PLOS ONE

Journal Requirements:

3. Thank you for stating the following financial disclosure: "DS is supported by the National Institute for Health and Care Research (NIHR) through an NIHR Advanced Fellowship (NIHR302955) and the NIHR Southampton Biomedical Research Centre (NIHR203319)."Please state what role the funders took in the study.  If the funders had no role, please state: ""The funders had no role in study design, data collection and analysis, decision to publish, or preparation of the manuscript."" 

4. Thank you for stating the following in the Acknowledgments Section of your manuscript: "DS is supported by the National Institute for Health and Care Research (NIHR) through an NIHR Advanced Fellowship (NIHR302955) and the NIHR Southampton Biomedical Research Centre (NIHR203319)"

Please remove any funding-related text from the manuscript and let us know how you would like to update your Funding Statement. Currently, your Funding Statement reads as follows: " Please state what role the funders took in the study.  If the funders had no role, please state: ""The funders had no role in study design, data collection and analysis, decision to publish, or preparation of the manuscript."" 

Additional Editor Comments:

Two reviewers and myself have read the submitted manuscript with interest but we do not think the paper would merit publication in its current form. However, we would be happy to consider a revised manuscript, provided that you were able to address the points raised by the reviewers. Particularly the methods and their description in the paper should be revised to clarify.

Reviewers' comments:

Reviewer's Responses to Questions

**Comments to the Author**

1. Is the manuscript technically sound, and do the data support the conclusions?

Reviewer #1: Yes

Reviewer #2: Yes

2. Has the statistical analysis been performed appropriately and rigorously? 

Reviewer #1: Yes

Reviewer #2: Yes

3. Have the authors made all data underlying the findings in their manuscript fully available?

Reviewer #1: No

Reviewer #2: Yes

4. Is the manuscript presented in an intelligible fashion and written in standard English?

Reviewer #1: No

Reviewer #2: Yes

5. Review Comments to the Author

Reviewer #1: The authors present an informative and timely investigation into the current state and trends in preconception health in the UK. In my view, descriptive studies like this, reporting on general population samples, are very important in guiding policymakers, for raising further, more complex research questions, and informing professionals in many fields. I think there is a strong rationale (intellectual, moral, and economic one) for improving children’s well-being by focusing on ‘upstream’ factors and by improving the health and well-being of the families they are born to. As mentioned by the authors, it is not only the children that benefit, and improvement in preconception health will benefit at least two generations. The key strength of the study has to do with using population-based studies, as other types of data sources might focus only on women or pregnant individuals, for example.

Introduction

Comment 1: The introduction section makes a strong case for the study. It highlights the impact that improving preconception health can have on multiple generations, and the authors seem to cover previous approaches to describing preconception health in the UK well. However, I wonder what the authors' rationale is for not describing findings from other countries? There are, of course, important differences in these phenomena and service systems even between European countries, but maybe there would be some previous findings from other countries that would be relevant for the current study? I appreciate that the authors might have an excellent rationale for adopting a UK-focused approach, but the rationale for that is not completely clear to me after reading the introduction section. I also think that the results will be able to inform professionals from areas outside of the UK.

Methods

Comment 2: I think the data in Table 1 would be best presented visually, or at least in a slightly different way. Although the authors have reported all relevant data in the table, currently it is somewhat difficult to read and to assess the overlap between data collection waves and the assessment ages across studies.

Comment 3: It seems to me that all three survey studies included individuals born in the UK. If this is the case, the implications of this need to be discussed clearly in the discussion section.

Comment 4: I think all in all the analysis approach is clear, and I appreciate the simplicity of it. If possible, I would prefer statistical testing that would provide estimates of the magnitude of the effect size, but of course, the magnitude of observed associations can be communicated in other ways too.

Comment 5: I think it would be important to report more information on the survey samples (e.g. missing data) for each survey in more detail than it has been done now.

Results

Comment 6: I find it somewhat hard to follow the results as they are being reported currently in the text. This might only be a matter of personal preferences, but I would consider doing some restructuring, and not reporting on each indicator separately. One thing that might simplify the results section would be that the descriptions of how the indicators were defined, and when information on each was available, would be described in a separate subsection of the results section, potentially in a table.

Comment 7: For some indicators (for example, for Ethnicity), it is not completely clear to me whether there was no significant difference found between ages/birth cohorts, or whether no statistical comparison was conducted. The reporting style could be made more clear. Otherwise the manuscript is very clearly written.

Discussion

Comment 8: I very much agree with the authors about the need for ongoing monitoring of health indicators from a public health point of view. I think the authors could describe in the discussion section what they think would be the best approach to this going forward, and what their recommendations are so that an even better monitoring can be done in the future.

Comment 9: Mental disorders are very much associated with one’s somatic health, health behaviors, ability to study and work, and importantly in the context of preconception health, they are associated with things such as the ability and willingness to take care of children, medications used before and during pregnancy, and breastfeeding. I think mental health is a very important dimension that should be considered when assessing preconception health. I appreciate that including these in the current study was not possible because of the limitations of the datasets, but I do think mentioning these factors would be important in the discussion section. Again, I don’t think this is necessarily a weakness of the current study, but given that the aim is to improve preconception health and our understanding of the current state of it, I think the importance of also assessing mental disorders could be mentioned here too.

Reviewer #2: Thanks for the opportunity to review this interesting paper which is really well written.

I have a couple of questions and suggestions I would be grateful if you could clarify please.

You reported using cohort sweeps with comparable ages of participants of 16-17 and 25-26, but this meant excluding the most recent data from MCS, where the 2023 sweep captured data from 23 year olds. I would think 23 year olds are close enough in age to 25-26 year olds to include these most up to date data which would be informative for current policy. Please could you explain why you excluded the age 23 MCS sweep? (Presumably this post-school age MCS sweep also includes the education indicator too? And including it would enable MCS to be included in Table 3?) Ideally I would recommend that the data from MCS age 23 sweep be included in the paper which is why I have recommended a major revision, my other comments are minor.

I think when referring to differences between cohorts it would be clearer to say differences across cohorts instead of across sweeps. For example at line 220, I think "Among female participants, the proportion of those aged 16/17 who smoked was significantly different across sweeps" should read different across cohorts. This occurs later too e.g. line 235. The phrase different across sweeps could then be used for change within cohorts between different sweeps.

The sentence at line 327 is not entirely clearly worded: Moreover, increases of 33-44% of pregnancies complicated by type 1 diabetes and 90-111% of pregnancies complicated by type 2 diabetes have been reported in Northern Europe (30,31).

It would be helpful to understand the exact questions that were asked for indicators where you propose differences in interpretation could have influenced the results (e,.g categorisation of termination of pregnancy as a 'complication' or not.

Thanks for your time.

6. PLOS authors have the option to publish the peer review history of their article (what does this mean?). If published, this will include your full peer review and any attached files.

Reviewer #1: No

Reviewer #2: No

---

## [Author Response · Author response to Decision Letter 0]

24 May 2024

We have addressed and responded to all comments from the reviewers in the uploaded 'Response to reviewers' file.

---

## [Editor Report · Decision Letter 1]

13 Aug 2024

PONE-D-23-42108R1Preconception health in adolescence and adulthood across generations in the UK: findings from three British birth cohort studiesPLOS ONE

Dear Dr. Righton,

Thank you for submitting your manuscript to PLOS ONE. After careful consideration, we feel that it has merit but does not fully meet PLOS ONE’s publication criteria as it currently stands. Therefore, we invite you to submit a revised version of the manuscript that addresses the points raised during the review process.

We look forward to receiving your revised manuscript.

Kind regards,

Katriina Heikkila, PhD

Academic Editor

PLOS ONE

**Additional Editor Comments:**

Thank you for submitting a revised manuscript. You have addressed many of the reviewers' comments but some of the key feedback has been incompletely addressed. I have also noted a couple of additional comments on the overall presentation of the information in the manuscript.

1. As suggested by Reviewer 1, please describe in the Introduction-section, with references, what is known about this topic in other high income countries. I appreciate the study is about the UK but it is important to provide a clear context for the investigation, explicitly explaining what this study adds to what we already know anout this topic.

2. As suggested by Reviewer 2, please add analyses using data from the MCS age 23 years sweep. You cite a paper proposing that adolescence these days extends to age 24, but life course epidemiology strongly suggests that health and outcomes in early adulthood are comparable in the early to mid-20s. Further, given the limited sample sizes, particularly on some indicators, adding MSC data from the age 23 sweep would increase the value of the study considerably.

3. As a general comment, the interpretation of the findings needs to be toned down throughout the manuscript. In multiple places the authors write about their findings revealing or showing patterns. Given the observational nature of the studies included, the modest sample sizes, attrition and missing data, these types of statements on the findings are too strong. At the most, the evidence from these studies can suggest (not reveal) associations.

4. The authors provide the total numbers of participants in each study in the abstract but the numbers of individuals included in the analyses were, in fact, considerably smaller. The abstract should be revised to give the reader a realistic idea of study sizes, as this has important implications to the interpretation of the findings in terms of generalisability and analytical power.

---

## [Author Response · Author response to Decision Letter 1]

18 Oct 2024

Thank you for your feedback. We have added a paragraph to the Introduction section, providing an overview of what is known about this topic in other high-income countries, with relevant references (lines 39 – 53). We have also included this evidence for comparison in the discussion (lines 307-310 and 329-332).

The MCS age 23 years sweep is currently ongoing, and data won’t be available until the end of 2025. We are therefore not able to add analysis using this sweep as part of the current paper. This has now been clarified in the methods section (lines 114-116). Please find more information here: https://cls.ucl.ac.uk/cls-studies/millennium-cohort-study/mcs-age-23-sweep/

We have carefully reviewed the manuscript and replaced terms such as 'shows' and 'reveals' with 'observed' and 'noted' throughout the discussion to better reflect the observational nature of the studies and avoid overstating the findings. 

We have revised the abstract and added the sentence: 'However, data for these indicators were not consistently available for all cohort members’, to address concerns about generalisability and analytical power.

---

## [Editor Report · Decision Letter 2]

5 Nov 2024

Preconception health in adolescence and adulthood across generations in the UK: findings from three British birth cohort studies

PONE-D-23-42108R2

Dear Dr. Righton,

We’re pleased to inform you that your manuscript has been judged scientifically suitable for publication and will be formally accepted for publication once it meets all outstanding technical requirements.

Kind regards,

Katriina Heikkila, PhD

Academic Editor

PLOS ONE
---

## [Editor Report · Acceptance letter]

7 Nov 2024

PONE-D-23-42108R2 

PLOS ONE

Dear Dr. Righton, 

I'm pleased to inform you that your manuscript has been deemed suitable for publication in PLOS ONE. Congratulations! Your manuscript is now being handed over to our production team.

Kind regards, 

on behalf of

Dr. Katriina Heikkila 

Academic Editor

PLOS ONE